# Enhanced Ferroelectric and Dielectric Properties of Niobium-Doped Lead-Free Piezoceramics

**DOI:** 10.3390/ma16020477

**Published:** 2023-01-04

**Authors:** Faysal Naeem, Mohsin Saleem, Hamid Jabbar, Gulraiz Tanvir, Fiza Asif, Abrar H. Baluch, Muhammad Irfan, Abdul Ghaffar, Adnan Maqbool, Tayyab Rafiq

**Affiliations:** 1School of Chemical and Materials Engineering (SCME), National University of Sciences and Technology (NUST), Islamabad 44000, Pakistan; 2School of Interdisciplinary Engineering & Sciences, National University of Sciences and Technology (NUST), Islamabad 44000, Pakistan; 3Department of Mechatronics Engineering, College of Electrical and Mechanical Engineering (CEME), National University of Sciences and Technology (NUST), Islamabad 44000, Pakistan; 4Department of Material Science & Engineering, Institute of Space Technology, Islamabad 44000, Pakistan; 5Department of Physics, Government College University, Lahore 54000, Pakistan; 6Department of Metallurgical & Materials Engineering, University of Engineering & Technology (UET), Lahore 54890, Pakistan

**Keywords:** piezoelectric, dopant, ferroelectric, dielectric, materials

## Abstract

Lead-free ceramics are promising candidates for replacing lead-based piezoelectric materials such as lead-zirconate-titanate (PZT) if they can compete in dielectric and ferroelectric characteristics. In this work, for lead-free piezoelectric ceramic, 0.74(Bi_0.5_Na_0.5_TiO_3_)-0.26(SrTiO_3_) (BNT-ST26) and niobium-substituted (Nb-BNT–ST26) ceramics were synthesized by solid-state reactions. The evolution of niobium substitution to the perovskite phase structure of BNT-ST26 ceramics was confirmed by X-ray diffraction (XRD) analysis and Raman spectra. Electromechanical properties of Nb-BNT-ST26 ceramics initially increased with the addition of niobium up to 0.5% and decreased with a further increase in Nb content. Temperature-dependent dielectric curves showed that the depolarization temperature (T_d_) decreased below room temperature because of Nb substitution. The composition with 0.5% Nb yielded a maximum bipolar strain (S_max_) of 0.265% and normalized strain of d_33_* ~ 576 pm/V under an electric field of 4.6 kV/mm at room temperature. At this critical concentration of 0.5% Nb, maximum saturation polarization of 26 μC/cm^2^ was achieved. The dielectric constant with temperature peaks became more diffused and the depolarization temperature decreased with the increasing Nb content. The study concludes that Nb-doped BNT-ST26 is an excellent material for high-temperature, stable, frequency-dependent, lead-free piezoelectric devices.

## 1. Introduction

Lead-based ceramic materials such as lead-zirconate-titanate (PZT) have dominated the piezoelectric market because of their ease of fabrication, low cost, and outstanding ferroelectric and dielectric properties with high curie temperatures of 390 °C [1]. The PbTiO_3_-PbZrO_3_ system is the most notable piezoelectric material for actuators, transducers, and sensors [1,2,3]. The high concentration of lead (>50%) and the toxicity of highly volatile PbO during sintering made these materials unsafe for the environment and human health. The increasing restrictions on the use of lead for environmental and human health protection promoted the research and development of safe and sustainable materials and products [4,5,6]. Consequently, lead-free materials have gained attraction for environment-friendly piezoelectric applications.

Among the lead-free piezoelectric ceramics, bismuth-based BNT-ST is one of the most promising candidates for replacing lead-based materials because of its high strain characteristics [4]. The solid solution of (Bi_0.5_Na_0.5_)TiO_3_ exhibits morphotropic phase boundaries (MPB) and has a rhombohedral perovskite structure belonging to the *R3c* space group [5]. Although it has high remnant polarization (P_r_) and strain values, its functionality as an actuator is limited because of the high coercive field of 7 kV/mm [6,7]. One way to overcome this problem is to modify it with other perovskite materials such as SrTiO_3_, showing high strain values, suggesting potential application as an actuator. This can further be optimized by the introduction of dopants into A or B sites of the perovskite structure or binary or tertiary solutions with other perovskite structures [7,8]. SrTiO_3_ has a perovskite structure belonging to the space group Pm3m [9]. Its solid solution with BNT-ST is reported by a few papers, showing promising ferroelectric characteristics [10,11,12,13,14,15,16,17,18].

Our work reports the detailed investigation of the 0.74(Bi_0.5_Na_0.5_TiO_3_)-0.26(SrTiO_3_) (BNT-ST26) ceramic prepared by the conventional solid-state reaction method. The effect of niobium doping on the BNT-ST26 perovskite structure was investigated by analyzing phase analysis, microstructure, dielectric, and ferroelectric properties. The thermal stability of the ceramics was studied by the polarization loops at different temperatures. The dielectric constant as a function of temperature, frequency, polarization, and strain loops is discussed in detail in this report.

## 2. Synthesis of BNT-ST26 Powder

### 2.1. Powder Preparation

The piezoelectric ceramic was synthesized by the traditional solid-state reaction method, which uses reagent-grade oxides and carbonates. All the reagents, bismuth (III) oxide (Bi_2_O_3_; 99.975%), sodium carbonate (Na_2_CO_3_; 99.9%), titanium dioxide (TiO_2_; 99.9%), niobium pent-oxide (Nb_2_O_5_; 99%), and strontium carbonate (SrCO_3_; 99%), were measured following the stoichiometric formula (1 − x) (0.74(Bi_0.5_Na_0.5_TiO_3_) − 0.26(SrTiO_3_)) – xNb. A total of 5 compositions were selected with varying concentrations of niobium (x = 0.0%, 0.5%, 1.0%, 2.0%, and 5.0%). The measured reagents were ball-milled with ethanol for 24 h and dried for 12 h at 90 °C. The resultant powder was then calcined at 850 °C for two hours followed by regrinding with ethanol for 24 h and drying for 12 h at 90 °C. The dried powder was mixed with polyvinyl alcohol (PVA) binder to make compact pallets (5-ton pressure, 12 mm diameter). Compacted pellets were sintered for 4 h at 1150 °C. The silver paste was then applied to the sintered pellets and heated for 30 min at 700 °C to make working electrodes.

### 2.2. Characterization Techniques

X-ray diffraction (XRD) (JEOL JDX-60PX) using a Cu-Kα radiation (λ = 1.5406 Å) source was carried out for structure analysis and confirmation of Nb doping. The microstructure and morphologies were investigated using a field emission scanning electron microscope (FE-SEM) (Hitachi, FE-SEM S-800, Tokyo, Japan). The Wayne Kerr Precision Impedance Analyzer (Bognor Regis, UK) was used to investigate the dependence of dielectric properties. Polarization–electric field (P–E) hysteresis loops and field-induced strains were measured using a piezoelectric evaluation system (AixaccT 2000, Aachen, Germany).

## 3. Results and Discussion

### 3.1. Structural Analysis

XRD patterns were recorded in the 2θ range of 20–80° (Figure 1) and were analyzed by the PANalytical-X’pert high-score program. The diffraction peaks of the powder were indexed according to the standard JCPDs card (No. 01-089-4934 and No. 00-036-0340). The analysis showed that the BNT part has an R (rhombohedral) structure, and the ST part has a C (cubic) structure, belonging to the space groups R3c and Pm3m, respectively. Splitting of the (111) peaks at about 2θ of 40.0° indicates R phase presence [13]. Furthermore, the peak shifting to higher angles was observed after the Nb additions, shown in the extended portion of Figure 1b centered at 2θ = 46.5° [12,13]. The shift may be due to the replacement of the smaller Ti^+4^ ion (ionic radius ~ 0.605 Å) with the larger Nb^+5^ ion (ionic radius ~ 0.64 Å) after doping [19]. All the peaks are indexed by a perovskite structure, suggesting that Nb is successfully dissolved into the lattice structure of the BNT-ST26 ceramic to make a homogeneous solid solution. This shows that all the ceramics were crystallized into a single-phase perovskite structure, having no impurity peaks [13,20]. The 5% Nb shows a small impurity peak appearing at an angle of 28° which is due to un-substituted niobium in BNT-ST26 ceramics.

#### 3.1.1. Microstructure

SEM analysis showed dense microstructures for all the samples, as shown in Figure 2. The substitution of Nb influenced the average grain size, and the grain morphology of the prepared samples to some extent. The increasing concentration of Nb resulted in the decrease of average grain from 1.7 μm at 0% Nb to 0.4 μm at 5% Nb, as measured by the line intercept method. The surface morphology showed that the relatively sharp corners in grains of non-doped BNT-ST26 ceramic converted to relatively round ones after Nb additions. Additionally, a particularly dense morphology with a more uniform surface was observed at higher Nb concentrations [19,21]. Hence, the increasing Nb concentration affected the grain size and to some extent the surface morphology. The difference in the ionic radii of Ti^+4^ and Nb^+5^, at 0.605 and 0.64 Å, respectively, may have influenced the variations in grain morphology and microstructure after doping [19,21,22,23]. Another reason for these results may be because of the reaction between titanium (Ti) and niobium (Nb) that can suppress grain growth.

#### 3.1.2. Raman

The Raman spectra showed several modes for the doped and undoped BNT-ST26 ceramics in the range of 100–1000 cm^−1^, as shown in Figure 3. This wavenumber range was further divided into four modes. The first mode at around 180 cm^−1^ is linked to the movement of the A-site in the perovskite structure [24]. This can either be because of the clusters of BiO_6_ and NaO_6_ octahedra or cation distortion in the structure [24]. No change in the first mode was observed before and after doping. The shifts in II, III, and IV modes around 270, 500–600, and 800 cm^−1^, respectively, indicated that niobium is partially substituted for titanium in BNT-ST26 ceramics, resulting in the distortion of TiO_6_ at the B-site. The peak shifted from lower wavenumbers to higher ones due to the B-site distortion with increasing Nb content. Ti^+4^ with an atomic radius of 0.605 Å was substituted with Nb^+5^ having an atomic radius of 0.64 Å, causing oxygen vacancies, which result in atomic asymmetry [24,25,26]. Increasing Nb content results in more distortion, which decreases the ferroelectric behavior, as confirmed by further testing.

### 3.2. Dielectric Studies as a Function of Temperature

The dielectric constant (ε_r_) with respect to temperature charts of Nb-doped BNT-ST26 at frequencies of 1, 10, and 100 kHz is shown in Figure 4. After Nb 0.5 addition, the peak ε_r_ value was observed at 4450. The low loss tangent (tanδ) corresponding to T_m_ is 0.029 at 1 kHz, as shown in Table 1. The peak broadening was observed with the initial Nb addition, with peaks becoming narrow at higher concentrations. The peak also shifted to the lower-temperature zone with increasing Nb (Figure 4). The high diffuseness shown by dielectric permittivity indicates that ceramic behaves like a relaxor ferroelectric, which was confirmed by other reports [17,18]. Relaxor ferroelectrics are ferroelectrics that have inhomogeneous composition, and their curie point T_c_ is not well-defined. Owing to the structural inhomogeneity, all domains of the ceramic do not depolarize at the same temperature [17]. Therefore, depolarization temperature, T_d_, or relaxor to FE (ferroelectric) transition temperature, T_R–F_ (relaxor to ferroelectric), points were not noticed after Nb substitution. This can be considered as a combined state of the FE phase (ferroelectric) at lower temperatures and the relaxor phase at high temperatures [18,27,28,29,30]. In the high-temperature range, as the temperature, T_m_, the temperature corresponding to the E_max_, the dielectric permittivity diverges from Curie–Weiss law.

#### 3.2.1. Ferroelectric Characterizations

##### Polarization vs. Electric Field

The shape and size of the measured P–E curves of the doped and undoped ceramics were studied to reveal the ferroelectric properties of these materials under an electric field, as shown in Figure 5. P_r_ and E_c_ (remnant polarization and coercive field, respectively) continuously decreased with increasing Nb contents and temperatures (Figure 5d). These results indicate the transition of ferroelectric behavior from normal to relaxor. The BNT-ST26 is the ergodic relaxor ferroelectric, due to the presence of fluctuating polar nano-regions (PNRs) [12,31,32]. The polarization hysteresis loop has a distinct sharp rise above coercive field E_c_ and low P_r_ values at zero electric fields. Resultantly, PNRs cannot preserve polarization when the applied electric field approaches zero. The gradual decay in the polarization (ferroelectric characteristics) at elevated temperature further reinstates the ergodic relaxor behavior of the polarization curves. A similar frequency dispersion response at T_m_ was observed in all samples.

A narrower loop was seen at 50 °C and higher temperatures for ceramics with 5% Nb, suggesting the instability of ferroelectric (FE) at these temperatures, as seen in the Appendix A. However, relatively wider loops were observed for non-doped ceramics at all temperatures, particularly at room temperature, showing relatively higher temperature stabilities of these non-doped ceramics compared to the doped ones. With increasing temperature, a decline in remnant polarization, P_r_, and coercive field, E_c_, values for niobium-substituted BNT-ST26 ceramics was observed. A polarization retention value beyond 75 °C is very small as compared to the P_r_ remanent polarization at room temperature, indicating that the instability of ferroelectrics is above 75 °C. This means that ferroelectric characteristics disappeared at high temperatures and the material behaved like conventional dielectric material [31,32,33].

##### Strain vs. Electric Field Curves

Bipolar butterfly strain curves of the non-doped and niobium-doped ceramics are shown in Figure 6a,b. The non-doped BNT-ST26 ceramic showed a strain (S%) of 0.22%. The equivalent normalized strain, d*_33_, was 478 pm/V, which is significantly greater than reported BNT-containing ceramics [34,35,36], and a small negative strain (S_neg_) of 0.019% was observed and the strain loop was found to be very saturated for non-doped ceramic (Figure 6a). After doping with 0.5% Nb, an increase in S_max_ from 0.22% to 0.265% and a drop of S_neg_ to ~0 were observed. The equivalent normalized strain, i.e., d*_33_, was increased from 478 to 576 pm/V (a 19% increase) with just 0.5% Nb addition to non-doped BNT-ST26 ceramic, as presented in Figure 6b. The drop in behavior indicates the existence of the ferroelectric (FE) phase and relaxor to ferroelectric (RFE) phases. This behavior suggests that the Nb^+5^ addition disrupts the polarization, resulting in slim and symmetric loops, and subsequently, ceramic behaved like relaxors [37,38]. The reduction in S_neg_ is because the ferroelectric material transitioned towards an ergodic relaxor. However, with a further increase in Nb concentration, the strain values significantly decreased. This behavior is in accordance with the other characterizations that showed the degradation of FE with increasing concentration. The change in the size and shape of strain curves due to increasing Nb concentrations implies the existence of electric field-induced relaxor phase formation [38].

The unipolar strain curves recorded at room temperature for the BNT-ST system are presented in Figure 7. The size and shape of S–E loops were changed with the increasing Nb concentration, suggesting the existence of an electric field-induced relaxor phase [12,34,39,40]. The 0.5% Nb-doped BNT-ST26 ceramic showed the maximum unipolar strain value of 0.23%. Furthermore, the strain curves were symmetric and the negative strain (S_neg_) was not observed, suggesting that a normal field formed coexisting FE and relaxor patterns. Polarization results also support the field-created strain behavior, as demonstrated by the shape and construction of the P–E loops (Figure 5) and Table 2.

The normalized strain, d*_33_, as a function of Nb concentrations is shown in Figure 7b. This strain shows the change in volume of a piezoelectric material under an applied electric field. It is calculated by unipolar strain/electric field at 4.7 kV/mm. Niobium-substituted BNT-ST26 with a concentration of 0.5% Nb displayed a high S_max_ of 0.225% and d*_33_ of ~480 pm/V. A further increase in the concentrations of Nb resulted in decreased normalized strain values.

To understand the electric field-induced strain, the strain vs. the square of polarization (S–P^2^) relation was calculated and shown in Figure 8. The strain varied proportionally to the square of the polarization. Undoped BNT-ST26 ceramic showed a parabolic curve having small hysteresis, whereas the Nb doping decreased the hysteresis, which indicates the exertion of electro-strictive strain. The Nb doping changed the strain from piezoelectric for pure BNT-ST26 to electro-strictive strain.

## 4. Conclusions

In this work, BNT-ST26 ceramics were successfully synthesized by solid-state reaction. The XRD and Raman analyses showed that the niobium was successfully substituted in the BNT-ST26 perovskite matrix. A detailed investigation of dielectric, ferroelectric, and strain characteristics with temperature was performed, which showed that niobium substitution increased the electromechanical properties until the optimum concentration of Nb 0.5%, with further increases in concentration significantly decreasing the properties. The critical composition of 0.5% Nb in BNT-ST26 resulted in a high P_r_ = 2.62 μC/cm^2^, bipolar S_max_ = 0.265% at E = 4.6 kV/mm, and d*_33_ = 576 ρm/V. Normalized strain at 0.5% Nb showed a 19% increase in non-doped BNT-ST26 ceramic. The temperature-dependent FE characteristics after Nb addition showed that the FE and relaxor phases co-existed in the broader temperature range with the reduction of S_neg_, and maximum saturation polarization, compared to non-doped BNT-ST26. This study concludes that Nb 0.5% BNT-ST26 is a promising candidate for a lead-free ceramic material for piezoelectric applications. 

## Figures and Tables

**Figure 1 materials-16-00477-f001:**
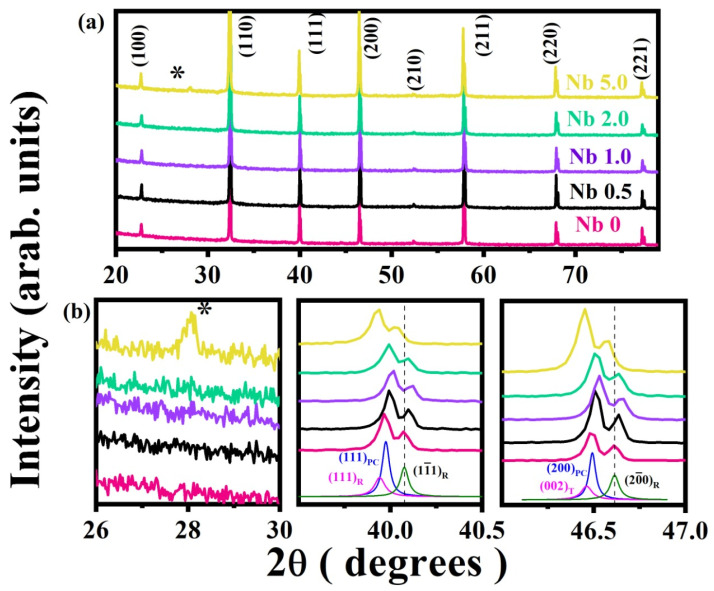
(**a**) XRD patterns of Nb-doped BNT-ST26 ceramics, and (**b**) magnified view of peak evolution around 2θ of 28°, 40°, and 46.5°. * impurities at Nb 5.0.

**Figure 2 materials-16-00477-f002:**
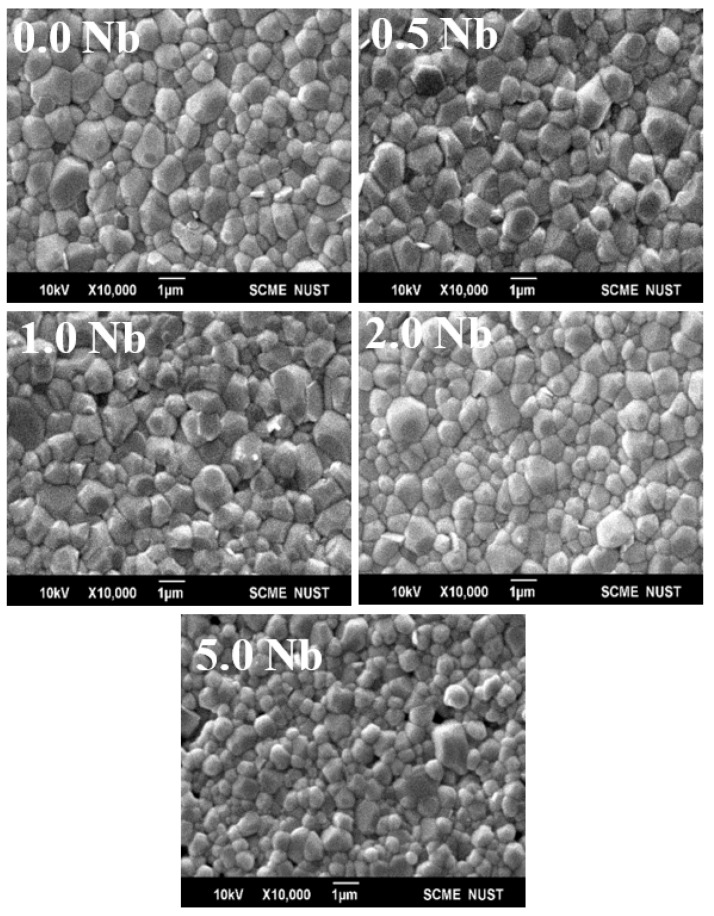
SEM images of pure and Nb-doped BNT-ST26.

**Figure 3 materials-16-00477-f003:**
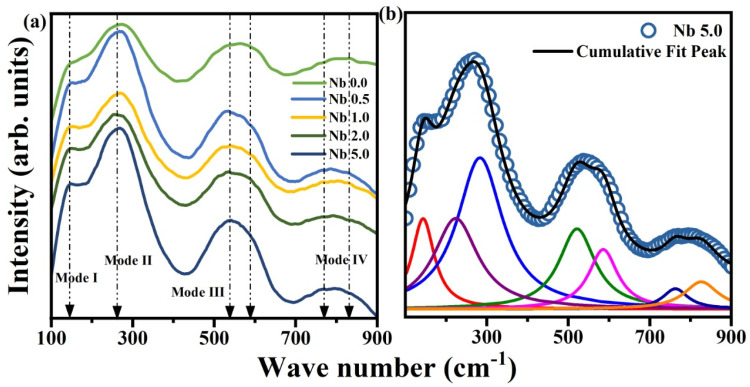
(**a**) Raman spectra of pure and Nb-doped BNT-ST26. (**b**) Raman spectrum of Nb-doped ceramics deconvoluted in accordance with Lorentzian functions.

**Figure 4 materials-16-00477-f004:**
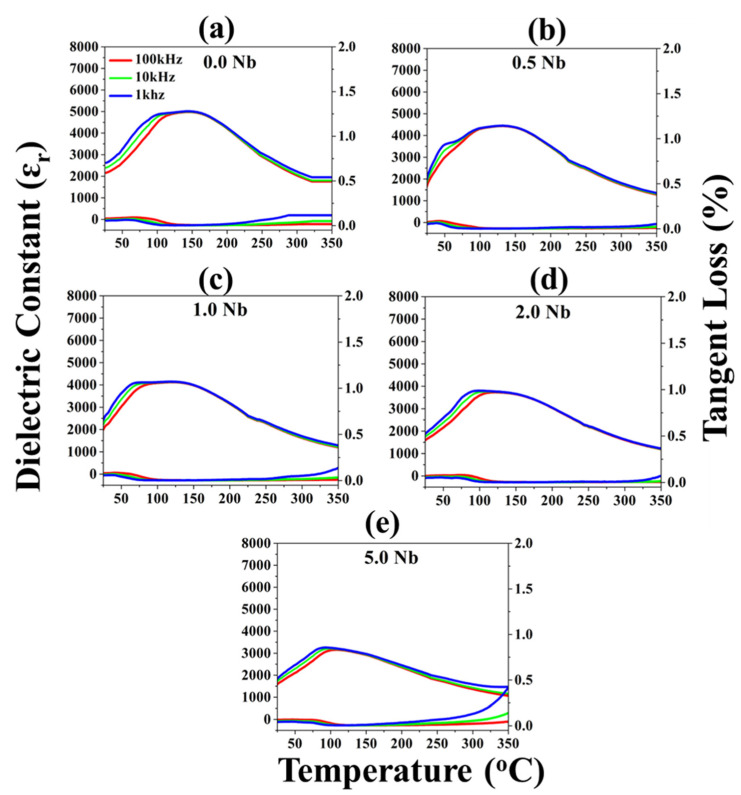
Dielectric constant vs. temperature curves of non-doped (**a**) and Nb-doped BNT-ST26 with (**b**) 0.5% Nb, (**c**) 1.0% Nb, (**d**) 2.0% Nb, and (**e**) 5.0% Nb.

**Figure 5 materials-16-00477-f005:**
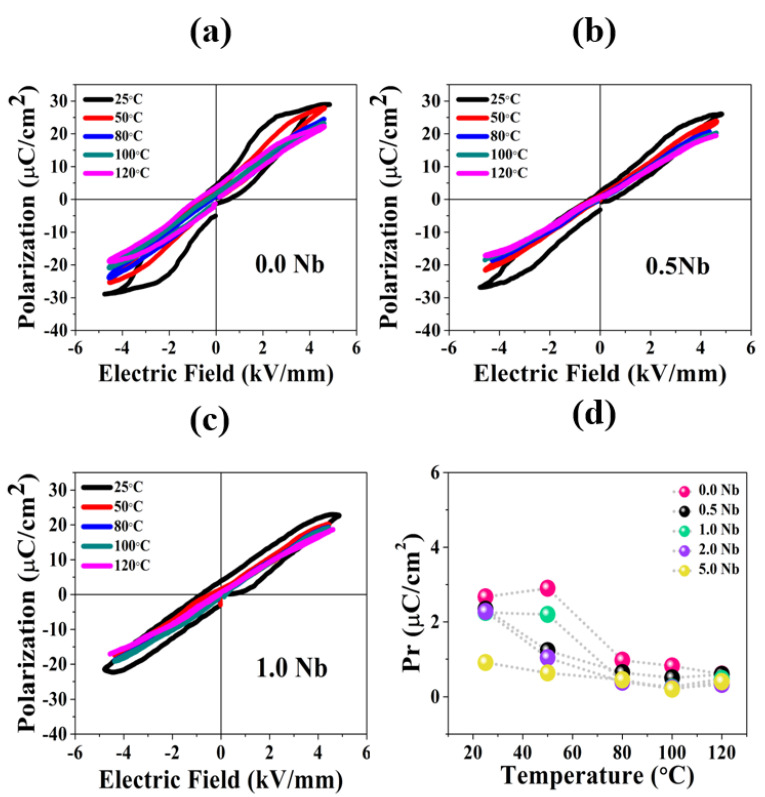
Polarization vs. electric field graphs of (**a**) non-doped, (**b**) Nb 0.5%-, (**c**) and Nb 0.1%-doped BNT-ST26. (**d**) Polarization retention, P_r_, vs. temperature of all compositions.

**Figure 6 materials-16-00477-f006:**
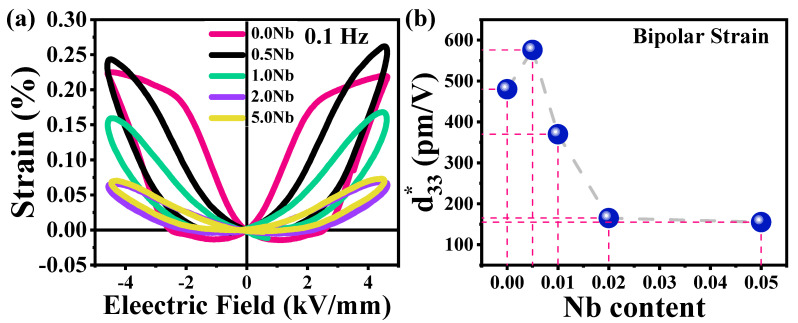
(**a**) Bipolar S–E curve of non-doped and Nb-doped BNT-ST26, and (**b**) d*_33_ values calculated by bipolar strain curves at all concentrations.

**Figure 7 materials-16-00477-f007:**
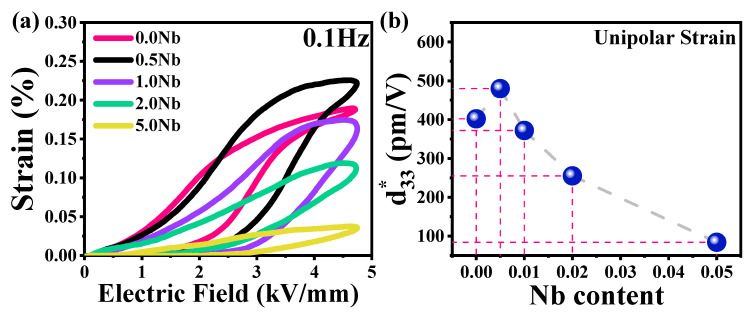
(**a**) Unipolar S–E curve of pure and Nb-doped BNT-ST26, and (**b**) d*_33_ values calculated by unipolar strain curves at all concentrations.

**Figure 8 materials-16-00477-f008:**
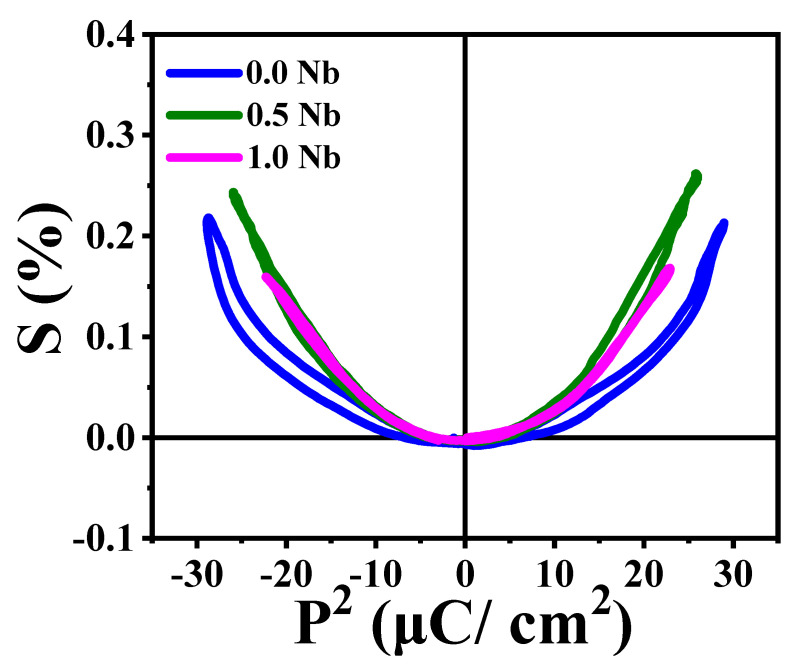
S–P^2^ curve of undoped and doped BNT-ST26 ceramic.

**Table 1 materials-16-00477-t001:** Calculated parameters from dielectric, polarization, and strain curves.

Nb %	ε_r_ 1 kHz	tanδ 1 kHz	T_m_ (°C)	P_r_ (µC/cm^2^)	P_s_ (µC/cm^2^)	S_max_ (%)	d*_33_ ρm/V	Density g/cm^3^
0.0	5009.4	0.0041	140	2.67	28.9	0.22	480	5.5
0.5	4450.3	0.0029	137	2.34	26.0	0.265	576	5.7
1.0	4144.6	0.0030	123	2.25	22.6	0.17	370	5.7
2.0	3804.4	0.0086	109	2.28	18.3	0.075	165	5.6
5.0	3257.2	0.0096	101	0.91	13.1	0.07	155	5.6

**Table 2 materials-16-00477-t002:** Comparison of maximum strain results of different BNT-based ceramics.

Composition	E_max_	S_max_	d*_33_ (pm/V)	References
BNT-ST26–Nb	4.6	0.265%	576	Our Work
BNT–ST26	4.6	0.22%	480	Our Work
BNT–ST24	7.0	0.20%	266	2020 [41]
BNT–ST25	4.0	0.28%	600	2014 [31]
BNT–ST28	6.0	0.29%	488	2008 [34]
BNT–BT–KNN	8.0	0.45%	560	2007 [8]
BNT–BKT–Nb	5.0	0.21%	420	2019 [42]
BNT–BKT–Nb	7.0	0.448	641	2010 [22]
BNT–ST23–Fe	2.0	0.11%	550	2017 [43]
BNT–SZ–Nb	8.0	0.22	315	2015 [38]
BNT–BZ–Nb	6.0	0.17	283	2015 [39]
BNT–BT–SZ	4.0	0.235	590	2014 [24]
BNBT–SZ–Nb	3.0	0.20	830	2015 [12]
BNKT–ST–Ta	4.0	0.33	830	2016 [44]
BNKT–ST–LT	4.0	0.35	875	2022 [45]

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
