# Peer review of "Enhanced Ferroelectric and Dielectric Properties of Niobium-Doped Lead-Free Piezoceramics"

_materials, 2023, doi:10.3390/ma16020477_

Round 1

Reviewer 1 Report

The work refers to the topic of lead free piezoelectric ceramics of actual interest.

There are some issues that need to be addressed.

The phrase regarding Fig. 10 “It is observed that the impedance …at a frequency…ABC (Fig. 10)” is incomplete.

It is mentioned that d33* normalized strain is calculated at 5 kV/mm from the unipolar S-E curves (Fig. 14), but these curves do not reach 5 kV/mm.

The caption of Fig. 16 is “S-P2 curve”, but the graph shows S-P curves.

The English in some phrases has to be revised.

Reviewer 2 Report

This paper investigates the effect of Nb addition to non-lead BNT-ST ceramics. There are some interesting aspects, but there are also some problems that I would like to inquire about.

1. please describe what the dielectric constant, polarization, and strain values obtained in this study are in comparison to previous studies. The references cited are not up-to-date, so please look up newer papers and compare with them.

2. since the AC response has few findings obtained and there is a lot of duplication, it would be better to reduce the description of the AC response. Specifically, you may keep Figures 6 and 8 and delete the others.

3. it is our judgment that the strain curves are for relaxor ferroelectrics. Since a dimple is seen in the polarization characteristics and the rise of the strain curve is steep, it may be due to antiferroelectricity.

The following is a small problem.

4. outline line 11

ρm → pm

at → under

5. Abstract line 12

Please note the value of the electric field.

6. last line of Abstract

stable high temperature ? The paper doesn't say much about high temperature properties, and they don't seem particularly good.

7. page 2, line 14

polarization retention → remanent polarization

8. page 3, lines 12-13

as shown in figure 1: does not seem to be shown in the Fig. 1.

9. page 5, line 1

(Figure 2) → as shown in Figure 2

10. page 5, line 6

(Figure 2) : These expressions are found throughout the text, but many of them are unnecessary. Please delete them.

Figure 3 and Figure 4

In Figure 1, Nb content increases from the bottom, whereas in Figure 3 and Figure 4, Nb content increases from the top. Please align them in the order of Figure 1.

12. page 8, line 1.

Insert (εr) after dielectric constant.

13. Figure 5.

Axis Explanation Delete (εr) after Dielectric Constant.

Insert (%) after Tangent Loss.

Figure 6: Dielectric vs → Dielectric constant vs

14. Table 1

Please consider significant figures.

15. page 19, line 12

.5% → 0. 5%.

16. page 20, line 1

polarization → the square of polarization

17. figure 16 axis explanation

mC → μC

18. page 22, lines 7-8

Indicate the electric field of the Pr, Smax measurement.

19. references

Usually bold the volume, not the year of publication.

[35] Takenata → Takenaka

That is all.

Reviewer 3 Report

Comments on the manuscript " Enhancement of Electromechanical Properties of Niobium Doped Lead-free Piezoelectric Ceramic". In this paper, the authors explored the effect Nb on electromechanical properties of BNT lead free ceramics. The results are interesting and the authors show a detailed analysis of the properties of the material, however the manuscript has too many figures and redundant analysis that must be remove. Then the authors must ve remove Figure 4, 6, 7, 8, 9 10 and 11. Also, the analysis about this figures must to remove. In addition, in experimental section details about field omdiced strain equipment must be descript. In result and discussion section, the authors must be deconvolve the rhombohedral and cubic peaks on the angles between 45 to 49 and 27 to 29 degrees. Also, the authors must be deconvolve the modes II and III and show the shift. Finally, the conclusion must be improve.  

Reviewer 4 Report

Minor errors were found particularly at:

"The silver paste was then applied to the sintered pellets and heated for 30 mins at 700 oC to make working electrodes as shown in figure 1." Which statement does not correspond to the presented figure.

"minimum value is observed for 0.5wt% Nb-doped BNT-ST26 at frequency …ABC (Figure 10)." The frequency value is missing.

A few other typo errors were found: "remnant", "stain." 

Round 2

Reviewer 3 Report

The manuscript can be published in the present way. 

Author Response

1. Thank you for the revised manuscript. Overall, all reviewers' comments are satisfactorily addressed. Just one minor correction is requested : in the supplementary material, please make reference to the figures with the correct number (S1, S2 etc. instead of previous number - e.g., 4, 8 and so on).

A: Thank you for critical review

We have updated the supplementary data file